# The Motor and Leisure Time Conditioning of Young Table Tennis Players’ Physical Fitness

**DOI:** 10.3390/ijerph17165733

**Published:** 2020-08-08

**Authors:** Beata Pluta, Szymon Galas, Magdalena Krzykała, Marcin Andrzejewski

**Affiliations:** Faculty of Tourism and Recreation, Poznan University of Physical Education, Poland Królowej Jadwigi 27/39, 61–871 Poznań, Poland; galas.szymon@gmail.com (S.G.); krzykala@awf.poznan.pl (M.K.); andrzejewski@awf.poznan.pl (M.A.)

**Keywords:** leisure activities, young athletes, table tennis, motor fitness

## Abstract

The purpose of the study was to assess the association between physical fitness and the lifestyle determinants of elite junior table tennis players. The basic anthropometric characteristics (body height and body weight) were collected of 87 Polish table tennis players (girls, n = 38 and boys, n = 49, at different stages of sport training, targeted and specialized) aged 11–17 years. The level of special fitness tests from the Table Tennis Specific Battery Test were used, assessing reaction speed and displacement speed. All eight International Physical Fitness Test trials were also used to determine the level of general fitness of the participants. Selected questions from the Health Behaviour in School-Aged Children questionnaire were asked to measure factors associated with leisure time. The findings confirm a relationship between sedentary forms of leisure time activity and the training of young players at the targeted stage (*Z* = −2.93, *p* = 0.003 school days and *Z* = −2.12, *p* = 0.034 days off). Moreover, competitors with longer training experience more often chose active forms of spending free time. Knowledge of the global physical activity undertaken by young athletes during their leisure time provides a better understanding of their individual needs and may help young table tennis players to succeed at a world-class level in the future.

## 1. Introduction

Table tennis is a compound and technically difficult game because the player must act quickly, accurately, and in changing conditions. As Limoochi [1] and Faber et al. [2] observed, table tennis is regarded as an early entry sport and associations already try to find high-potential players at a young age. Table tennis performance itself at such a young age is influenced by individual differences in growth, maturation, training experiences, competition participation, and environmental factors (including leisure time). These factors may affect a player’s sport potential [3,4,5].

Several authors have studied the relationship between various anthropological variables and body composition for success in table tennis [6,7,8]. However, some research states that, between the ages of 10 and 14 years, no distinct anthropometric player profile exists [1,9].

Table tennis is characterized by highly developed motor skills, such as agility [10], reaction speed time [11,12], explosive power and strength [13], eye movement, and coordination [14]. According to Bandi [15], table tennis game skills include such traits as grip, attitude or playing position, and types of punches and leg movements.

In physiological terms, table tennis belongs to endurance- and speed-based disciplines with changing modes of effort and intensity. Due to the short, high-intensity periods under anaerobic metabolism that characterize a match, the most important physical capacities of table tennis players are endurance and velocity. However, strength, coordination, and agility may also play a key role in this sport [16,17,18]. In racket sports, especially in table tennis, the phosphagenic energy source is the main mechanism of resynthesize ATP (adenosine triphosphate) during a match, which is aided by the anaerobic glycolysis system, but only on efforts with higher duration (more than 15 s) [13]. The rally in table tennis normally takes 10 to 15 s, while a match ends in 10 to 25 min. This long duration consequently represents the aerobic system as principal output energy in a full match [9,16].

Literature addressing the characteristics of youth table tennis players is not extensive. Three objectives of performance analysis are relevant to the study of young table tennis players. Technique analysis is concerned with the mechanical aspects of technique and how skills translate into performance [4,19,20]. Tactical analysis is concerned with strategies and tactical decisions that are manifested in a game [21,22]. Physical aspects of performance can be determined through analysis and estimates of game intensity [23]. Other preliminary research has shown that motor ability skills, such as speed while dribbling, aiming at a target, ball skills, and hand-eye coordination seem to be promising ways to distinguish between talented and less talented young players [14,19,24,25].

However, an even more expansive array of methods can be used in performance analysis of child and youth table tennis players. An interesting direction of future research seems to be the study of the relationship between players’ special fitness and their leisure time interests. Such studies are particularly important for this age group of table tennis players. In this sport discipline, the number of hours of training per week is especially high at a young age [5]. Young table tennis players, especially representatives of the national youth team, are confronted with high physical, psychological, and social demands [26,27].

The available literature discussing young table tennis players has ordinarily focused on specific parameters viewed as relevant to the sport and recreation, separate to variation in the growth and maturity (including social maturity) characteristics of the players [28]. Although physical training in table tennis is an obviously important factor for sport success, there is a lack of evidence that, at the same level of training, young athletes who also have well-organized and diverse activities in their free time achieve the best results. Leisure activities awaken, deepen, direct, and consolidate the interests of children and young adults, and increased activity stimulates action and boosts motivation to show initiative [29,30]. Knowledge of the global physical activity undertaken by athletes during their leisure time and their preferences provides a better understanding of their individual needs and may help young table tennis players to succeed at a world-class level in the future.

In the scientific and methodological literature, numerous works have been devoted to the influence of social environment factors on the physical performance of children playing table tennis, and this has determined the relevance of this study. The main social factors related to the physical performance of children playing table tennis are family, peer and school support, self-determined motivation, coping, perceived stress, recovery experienced, leisure time activity (treating as a form of rest and relaxation), and adequate sports facilities [4,17,27]. Therefore, this study crucially aims to determine what factors influence the level of physical fitness of young table tennis players. To achieve this goal, our study was twofold, addressing both physical fitness and leisure activities of young top-level table tennis players. The purpose of this study was to assess the association between physical fitness and the lifestyle determinants (leisure time physical activities and sedentary behavior (SB)) of elite junior table tennis players (girls and boys, at different stages of sport training).

## 2. Materials and Methods

### 2.1. Participants

The study group consisted of young Polish table tennis players from the teams of two provinces: Dolnośląskie and Wielkopolskie (n = 87). They were born in 2002–2007 and trained in table tennis at the targeted or specialized sports training stage (according to the guidelines of the National Table Tennis Development Program in Poland for 2018–2033). The targeted training stage includes players aged 9–12 years, and the specialized training stage includes players aged 13–17 years. The study group was selected arbitrarily using the following criteria: Written consent from parents to participate in the research, membership in the province team, current license of the Polish Table Tennis Association, a minimum three-year training period, health conditions, allowing all physical fitness tests to be carried out, and playing style, requiring the use of rackets with a so-called smooth lining (excluding people using rackets with atypical cladding, such as anti-spin cladding, short pin, or long pin, where play is characterized by a different technique than topspin strokes used in a battery of special tests).

The study was conducted in compliance with the Declaration of Helsinki and was approved by the local ethics committee: The Bioethics Committee of the Karol Marcinkowski Medical University, Poznań, Poland (No. 543/18). All data were analyzed confidentially.

### 2.2. Procedures

Anthropometric measurements were taken, according to standard procedures, following the guidelines described by Martin and Saller [31]. Stature was measured to the nearest 0.1 cm with a stadiometer, with the child standing upright. Body mass was measured with a Tanita MC-780 scale (Japan) with GMON software (version 3.2.8). The age was calculated to the decimal using the date of birth (day, month, year) and the date (day, month, year) the anthropometric measurements were taken. Decimal age categories were based on one-year intervals (e.g., 12.50 to 13.49 = 13 years). Growth references for height, weight, and body mass index (BMI) were constructed with the lambda, mu, sigma (LMS) method, using data from a large, recent, population-representative sample of school-aged children and adolescents in Poland [32]. Participants were classified as underweight, normal weight, or overweight, according to age- and gender-specific cut-off points [33]. All anthropometric dimensions were measured on two occasions for a sample of 87 players to calculate intra-observer technical errors of measurement.

### 2.3. Special Motor Fitness

To determine the level of special fitness of young table tennis players, tests from the Table Tennis Specific Battery Test (TTSBT) were used, assessing reaction speed (T1, T2) and displacement speed (T3) [34]. In reaction speed tests, balls are thrown at high speed (70 balls/minute) to different areas of the table tennis table and the player has to perform forehand (T1) or backhand top-spins (T2). The tennis of balls that touch the table over 15 s is considered successful. In T3, balls are thrown at high speed (80 balls/minute) to the sides of the table and the player should alternately perform forehand and backhand top-spins. The tennis of balls that touch the table over 15 s is considered successful. The selected options included the use of two strokes, considered basic at the stage of targeted and specialized training, and top-spin forehand and topspin backhand, previously used for similarly aged participants [23]. The intraclass correlation coefficient (ICC) overall absolute agreement of the assessing parameters of the TTSBT was high (0.85). More specifically, the Cronbach’s α (coefficient of reliability or consistency) for T1 was 0.83, for T2 was 0.86, and for T3 was 0.69.

A Tibhar Robo Pro Junior (Germany), a compact machine with an oscillator and remote control, was used to carry out the test samples. For the test, the machine was programmed to throw balls without rotation and with a central ball spread angle. The balls were thrown in the following variants: (1) top-spin forehand diagonally, (2) top-spin backhand straight, and (3) mixed attempt to play once with top-spin forehand and once with top-spin backhand with the machine in the center of the table. Before the start of each test sample, the competitors performed a block of shaping exercises and special exercises and were informed about the correct way of performing the test samples. All tests were measured on two occasions to calculate intra-observer technical errors of measurement.

### 2.4. Motor Fitness

All eight International Physical Fitness Test (IPFT) trials were used [35] to determine the level of general fitness of the participants. The IPFT components were: the 600 or 800 meter (girls 12–19 years old) or 1000 meter (boys 12–19 years old) long run tests (LORs); the handgrip strength test, measuring the maximal strength of the upper limb using a calibrated and adjustable hand dynamometer (HGR - handgrip strength); the standing long jump test, measuring the lower limb power (SBJ - standing long jump); the bent arm hang (BAH) test, measuring the upper limb endurance strength; the sit-ups in 30 s (SUP) test, measuring muscular endurance of the trunk and hip flexors; running speed: 50 meters run (50 mR); the speed shuttle run (SHR) test of 4 × 10 meters, measuring speed and agility; and the sit and reach (SAR) test, measuring agility. All tests were carried out in one day. Assessment of the results of the movement tests was made using scoreboards, developed according to calendar age groups, separately for each gender. The classification standard for each of the individual IPFT trials is 50 points. The overall fitness (OF) index was used to assess the level of physical fitness. The athletes started the tests after they had performed standardized warm-up general activities (15 min). All players did eight trials in the same order. The standardization of the test parameters is captured in protocols, which include a detailed description of the materials, set-up, assignment, demonstration, training phase, testing phase, and registering of test scores.

### 2.5. Lifestyle

Selected questions from the Health Behaviour in School-Aged Children (HBSC) questionnaire were asked [36,37] to measure factors associated with leisure time. Cronbach’s α, a measure of internal reliability, was obtained for the scale, showing a reliability of the total of 0.88. Time spent in leisure time, specifically moderate to vigorous physical activity (MVPA) and vigorous physical activity (VPA), was measured using a Physical Activity Screening Measure [38]. Reliability was established at ICC = 0.77 and validity *r* = 0.40. This measure was used earlier in population studies in Poland [29,30]. Participants were asked to answer two questions. Q1: Over the past seven days, how many days were you physically active for a total of at least 60 min per day? Q2: Over a typical or usual week, how many days are you physically active for a total of at least 60 min per day? Prochaska et al. [38] found these questions to be reliable and to have acceptable validity in comparison with accelerometer data. The MVPA index was calculated based on the following formula: MVPA = (Q1 + Q2) /2, where MVPA = PA index, Q1 is the total of physically active days during the past seven days, and Q2 is the tennis of physically active days during a typical week.

The VPA index was calculated based on the question: outside of school hours, how often do you usually exercise to such an extent that you get out of breath or sweat? The response categories were “every day”, “4–6 times per week”, “2–3 times per week”, “once a week”, “once a month”, and “none at all”. This index has been found to have at least partial validity, as Booth et al. [39] found that adolescents who reported higher levels of VPA outperformed those with lower VPA on a running test. Children and youth aged 5–17 years should accumulate at least 60 min of MVPS daily. VPA should be incorporated, including that which strengthens muscle and bone, at least four times per week [35].

Leisure time spent on screen-based sedentary behaviors was assessed through three questions. SB1: About how many hours a day do you usually watch television (including videos) in your free time? SB2: Approximately how many hours a day do you play PC games or TV games (PlayStation, Xbox, Game-Cube, etc.) in your free time? SB3: Approximately how many hours per day do you use a PC for chatting online, surfing the internet, writing emails, homework, etc., in your free time? The following nine response options were the same for all three questions: “none at all”, “about half an hour a day”, “about 1 h a day”, “about 2 h a day”, “about 3 h a day”, “about 4 h a day”, “about 5 h a day”, “about 6 h a day”, and “about 7 or more hours a day”. Vereecken et al. [40] evaluated the test–retest reliability and relative validity of the questions, assessing screen-based sedentary behaviors, and found no systematic difference between test and retest (ICC = 0.76 for boys and ICC = 0.54 for girls).

All respondents were informed of the aim of the research and in which way the results would be used for scientific purposes and that their participation would be anonymous. Questionnaires were completed in whole groups during sports camp and took approximately 20 min to complete.

The list of test methods used to analyze the motor and leisure activity of young table tennis players is presented in Figure 1.

### 2.6. Data Analysis

The level of special fitness of young table tennis players was considered a dependent variable in this study. The independent variables were motor fitness and lifestyle determinants (which included a self-reported amount/level of PA and SB). Descriptive statistics were presented as mean and standard deviation or range (minimum–maximum). The difference in mean results between two independent groups was checked using a Student’s t-test for independent samples. The normality of the distributions was checked using the Kolmogorov–Smirnov test. In the event of failure to meet the Student’s t-test assumption about the normality of the distribution of the studied variable, and in the case of ordinal variables, the Mann–Whitney U test (*Z*) was used to check the significance of differences. Correlations between variables were checked using the Spearman’s rank correlation coefficient (*r_s_*). The level of significance was set at 0.05. IBM SPSS Statistics 23 (IBM Corp., Armonk, New York, NY, USA) was used for the statistical analyses.

## 3. Results

This study analyzed the data of 87 young table tennis players (age 13.4 ± 1.74 years). The sample characteristics and descriptive results are presented in Table 1. More than half of the players were 13 years old.

Based on BMI calculations (according to Cole, Lobstein) [31]), nearly 55% of the surveyed players displayed a proper level of nutrition of their body, while 27% were diagnosed as overweight and the remaining 18% as underweight. The analysis showed that boys had higher results than girls for BMI (according to Cole; Z = −2.66, *p* = 0.008).

The average OF index score of all participants was 424.1 ± 41.69 points. There were statistically significant differences between the results of girls and boys (*p* = 0.003), with higher test results obtained by girls. Such differences were also found for the majority of IPFT tests (*p* < 0.05), but no significant differences were found for the measurements of hand muscle strength (*p* = 0.839) or torso inclination (*p* = 0.06; Table 2).

There were no differences in the total point index between the groups separated by training stage (*p* = 0.077). The only differences were found for the standing long jump (SBJ) test (*t* = −2.08, *p* = 0.041) and the BAH test (*t* = −2.94, *p* = 0.004). In both cases, the higher scores were achieved by players at the specialist stage of training. Table 3 presents the results obtained by young table tennis players in three special fitness tests.

Statistical analysis did not show significant differences in the results of TTSBT trials between boys and girls. However, it showed that, in all test trials, significantly higher results were obtained by competitors at the specialist stage of training.

Young table tennis players, characterized by high general efficiency (high OF values), achieved higher results in T1 and T2. The relationship between the general fitness area and the special fitness area of young table tennis players applies only to selected test samples (Table 4). There were no significant differences related to the players’ genders or differences between the groups based on the stage of training (*p* = 0.081).

The average number of days per week allocated to PA by young table tennis players in their leisure time was 5.5 ± 1.62, while exercise at high intensity (VPA) was performed on average 4.1 ± 0.95 days per week. Statistical analysis did not show any significant differences between MVPA and VPA ratios by gender (Table 5). A statistically significantly higher VPA level was found among competitors at the specialist stage of training (*Z* = −3.13, *p* = 0.002). In addition, MVPA standards were met by 38% of young table tennis players, and VPA standards by 66%.

Boys used computer games on school days and holidays (SB2), statistically significantly more often than girls. No other statistically significant differences were found in leisure time activity (S1, S3) between groups by gender.

The use of a computers, tablets, or smartphones (SB3) on school days (*Z* = −2.93, *p* = 0.003) and days off (*Z* = −2.12, *p* = 0.034) was significantly more frequent among the respondents at the targeted stage of training.

The analysis of the collected data showed that the most common forms of PA undertaken in leisure time were walking and cycling. Over half of the respondents devoted at least 2–3 h a week to this activity. No statistically significant differences existed between girls and boys in their choice of leisure activities. The stage of sport training was also not crucial in choosing recreational forms of physical activity. The correlation analysis showed no statistically significant relationship between the special fitness area (TTSBT) and the leisure time activity of the table tennis players.

## 4. Discussion

This study was a novel attempt to analyze the relationships between selected fitness characteristics and some lifestyle determinants in a group of elite young table tennis players in Poland. In recent years, researchers have been increasingly interested in the correlations between various lifestyle determinants and the level of physical fitness among adolescent athletes [41,42]. However, no scientific reports exist on the relationship between environmental factors, such as leisure time activities and the level of general and special fitness among young table tennis players.

The data found in this exploratory correlational study suggests that there may be clusters of variables in three broad dimensions (motor fitness, special motor fitness, and lifestyle) that best represent them and therefore might be statistically organized under the dimensions for enhanced scientific quality for future research. Many authors have attempted to assess the physical fitness of young people practicing sport, including table tennis, treating it as an isolated variable, as exemplified by the research of Ulbricht et al. [43], Kramer et al. [44,45], and Faber et al. [5]. However, information from Table 4 suggests an overall correlation between motor fitness and special motor fitness, with a specific focus on some variables. The results obtained showed that Test 2 might be more discriminant or relevant than Test 1 and Test 3. In turn, Test 3 is the only test correlating with the SAR test, which makes sense given the nature of both tests.

The results presented in the study show that all the examined players presented an average level of general physical fitness (OF = 424.1 ± 41.69), estimated according to standards for Polish youth [35]. The results of other studies, including those found by Żak et al. [46] and Łubkowska and Troszczyński [47], show that athletes practicing other sports (e.g., badminton, Olympic taekwondo, or swimming), who are the same age as the examined table tennis players, achieved a similar OF value.

The higher level of OF of girls practicing table tennis, compared to their male peers, can be explained, among other factors, by their faster biological maturation at the investigated age [44,48,49]. A comparative analysis of the average results of girls and boys showed that not all IPFT results were above the average classification standards (Table 2). In the hand strength test, both girls and boys obtained results below norms. An analysis of the results of dynamometer measurement of hand strength of girls and boys reported that this test was insufficient to represent total strength at this stage of development [50].

The girls also obtained worse results in the test of flexibility. The low point indicator of the players in this sample is described in the literature on the subject [51,52]. However, the optimal level of flexibility is necessary in table tennis for the correct performance of technical elements [28]. Therefore, there is a need to increase the training measures that shape flexibility in the motor training process of these young players. Table tennis belongs to the group of sport disciplines for which the level of coordination and speed skills, especially of the upper limbs, should be at the highest level [53,54,55,56]. This study showed that the table tennis players obtained the best results in the SHR test, assessing the level of these abilities. These results are consistent with those obtained for players on the Polish youth teams of the same age in other sports, including swimming, Olympic taekwondo, football, acrobatics, boxing, judo, canoeing, cycling, basketball, athletics, handball, volleyball, fencing, triathlon, rowing, and free wrestling [47,57].

The assessment of the level of special physical fitness in young top-level table tennis players has not yet been widely studied [58]. Sport-specific technical skills are predominant factors, although a complex profile of physical performance factors is also required. For this reason, accurate and reliable methods of evaluation are very important for the training process. The review of literature revealed that most of the studies designed tests for speed or accuracy of basic skills, other aspects of physical fitness, or counterattack [23,59]. As determined by Kastikadelis [23], contemporary playing techniques in table tennis are very important for target precision counting and the accurate reflection of players’ strokes. The TTSBT is a reliable test battery that focuses on the evaluation of technical skills and the progress of table tennis performance of young players [21,23].

However, so far, no appropriate test standards have been developed for young tennis players based on their age; the only available standards are based on the gender of the players [34]. When analyzing the results obtained by the examined players in relation to the available standards, it should be noted that in T1 and T2, both girls and boys were classified as having a good and a very good level of special fitness, respectively. In T3, on the other hand, the level was basic and good [34]. The TTSBT results obtained by girls did not differ significantly from the boys’ results (Table 3). Perhaps this was because the individual level of differentiation of the respondents was low (high group homogeneity) and the sample size was limited. Similar relationships have been reported by Chillon et al. [60], among others. However, differences based on the stage of training were noticeable, probably because of the significantly higher volume and intensity of training [9,13].

The relationship between the tests characterizing the level of general and special physical fitness of young players is interesting. The results of T1 and T2, whose main purpose was to estimate each player’s capacity to react quickly to unpredictable throwing of balls were significantly connected with the upper (BAH) and lower (SBJ) body power of the players (boys and girls, at both stages of training) and with their running and speed abilities (SHR and R50 m). The results of the study correspond with the findings of Kramer et al. [44] and Faber et al. [5], who studied junior elite tennis players in the Netherlands. In turn, the results of T3 were connected with upper body power (BAH) and body agility (SAR) of young players (Table 4). This was due to the characteristics of the test sample, with the main purpose to estimate the players’ capacity to perform lateral, lateral with pivot, profundity, and mixed displacements during a short period. A possible reason for these findings is that agility, speed, and strength of upper limbs includes dynamic movements requiring high muscle power, so one would therefore expect these performances to be closely related. Another explanation for the close correlation between these skills may be the same energy systems that each movement type demands [60].

One of the main goals of this study was to identify young table tennis players’ leisure time physical activities and SB between training. The level of PA (MVPA and VPA) and sedentary lifestyle (SB) outside the training of young athletes will allow trainers to understand the interests of their players better and may affect the individualization of the training process. According to Exel [61], it is difficult for young athletes and trainers to determine the best choices of leisure time strategy, but individualization of needs and a context-based approach in daily life schedule structure seems to be critical. For young athletes, considering table tennis performance, PA and SB profiles should also be considered key to evaluating, preventing, and treating overtraining symptoms [62]. The level of PA and SB between training may also highlight issues related to the health parameters of young athletes. PA at moderate to vigorous intensity can help prevent weight gain and maintain a healthy body weight in children and adolescents [63]. Current PA guidelines for children and youth aged 5–17 years recommend at least 60 min of MVPA per day and VPA at least four times per week [64], but nationally, only a small proportion of youth meet these guidelines [65]. Moreover, more than two thirds of European youth can be categorized as insufficiently active [61,66]. In our study, MVPA standards were met by 38% of young table tennis players, and VPA standards by 66% (Table 5). Although young athletes mostly performed MVPA above the recommended level and were also active in their leisure time, sedentary lifestyle habits associated with daily activities and social behaviors of young people were also visible. According to Pearson et al. [67], young elite athletes prefer to sit down as often as possible to aid recovery and prevent fatigue. This suggests that some active children choose to spend considerable time being sedentary (TV watching, cell phone interaction, recreational computer use, or other forms of screen-based entertainment), as confirmed by the research of de Rezende et al. [68]. Thus, it can be assumed that the studied young elite table tennis players, exposed to many hours of training and competition, more often adopt a sedentary lifestyle, treating it as a form of rest and relaxation. This is also reflected in the forms of PA undertaken by the examined table tennis players (girls and boys) in their leisure time (walking and cycling), which are characterized by low physical effort.

The results of this study do not completely confirm the assumption of a difference in terms of gender and stage of sport training in the leisure time activities of young table tennis players. Physical inactivity and sedentary behavior of young people are pervasive and persistent public health challenges to overcome. The results obtained on the differences between the two groups separated by training stage reflect typical cohorts, in which PA decreases and sedentary behavior increases. Many authors showed that it was determined that increased sedentary time was associated with age and negative health outcomes in both boys and girls [40,42,65,66,67]. A knowledge of how young people use their time could be instrumental in informing health interventions, modeling their behaviors, and planning physical activities.

The main limitation of this study that should be highlighted is that it assessed only young Polish table tennis players, and further studies should be performed with athletes of different nationalities. The relatively small group sizes present another limitation. The current study did not examine other potentially influential factors (such as differences in maturation, which might influence the physical fitness results, family leisure time and sport traditions, motivation, parental and peer support, training facilities, and school policies), which might also be considered a limitation. However, the similar cultural backgrounds of the examined players (coming from urban areas of cities) may be an advantage.

## 5. Conclusions

The results presented an idea of an under-explored way of combining important variables in the context of child and youth sport, namely assessment of the level of physical fitness and the quality of free time use between training.

The results confirm only the relationship between sedentary forms of leisure time activity (screen-based entertainment (SB3)) and the training stage of young players. This relationship occurred only in the group of players at the targeted stage of training. Although no significant relationships were found at the specialist stage of training, the fact that competitors with longer training experience more often chose active forms of spending free time (such as cycling and jogging) is noteworthy. Although this was not studied, we believe it fits in very well with the training process, being individually implemented training with the character of active regeneration, with a low intensity level. These types of activities, in addition to the basic variables, such as proper nutrition, hydration, and sleep, can contribute to achieving full readiness for the physical loads implemented the next day in specialist training of young table tennis players.

Because proper recovery is essential for the quality of training, the reported sedentary lifestyle of elite young table tennis players should not be a problem. This information may provide useful data for trainers in helping them control the loads that may influence training performance, as well as the health condition of young table tennis players. Coaches should suggest low intensity and pleasurable PA as a form or breaking the sport-specific intense routine, maintaining the fitness levels, and optimizing recovery and injury prevention through active recovery, while counselling them on how to use sedentary behavior for the benefit of table tennis players’ health, wellbeing, and performance. The research issues presented in this paper prove the necessity of conducting an analysis of this type through larger-scale studies.

## Figures and Tables

**Figure 1 ijerph-17-05733-f001:**
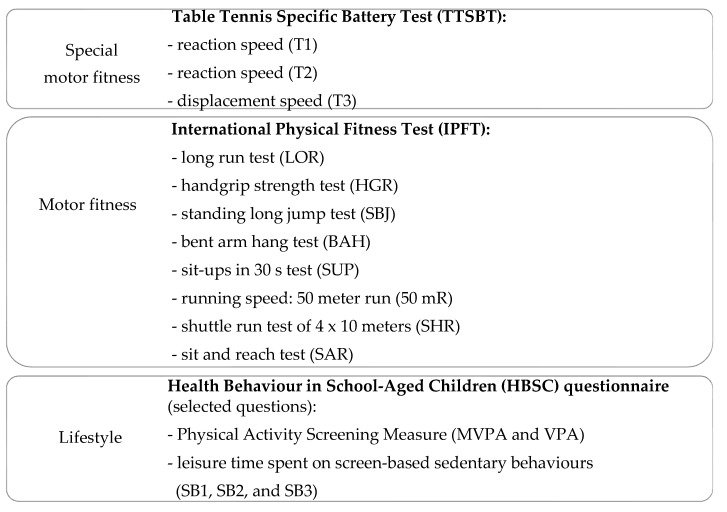
Recommended tests to gauge motor and leisure activity of young tennis players.

**Table 1 ijerph-17-05733-t001:** Demographic characteristics of the players.

Variable.	Girls (n = 38, 43.7%)		Boys (n = 49, 56.3%)	
	M ± SD	Min–Max	M ± SD	Min–Max
age (years)	13.4 ± 1.76	10.7–16.5	13.7 ± 1.75	11.0–17.0
body weight (kg)	47.2 ± 8.97	30.1–63.2	55.1 ± 14.15	28.4–92.2
body height (cm)	158.2 ± 8.99	141.0–178.5	165.0 ± 11.81	138.0–189.9
BMI (body mass index)	98.0 ± 12.61	80.1–145.1	106.2 ± 16.15	73.5–137.2
targeted stage of sport training (n, %)	23, 60.5%		27, 55.1%	
specialized stage of sport training (n, %)	15, 39.5%		22, 44.9%	

M = mean, SD = standard deviation, Min. = minimum value, Max. = maximum value.

**Table 2 ijerph-17-05733-t002:** Results of the International Physical Fitness Test (IPFT): comparison by gender.

Variable	Boys		Girls		Test t Student or U	
	M	SD	M	SD	*t/*Z	*p*
OF [pts]	412.5	43.75	439.1	33.87	−3.10 ^*^	0.003
50 mR [pts]	55.2	8.30	58.7	6.44	−2.14 ^*^	0.036
SBJ [pts]	52.4	7.97	58.2	8.18	−3.30 ^*^	0.001
LOR [pts]	50.8	6.22	54.5	5.76	−2.81 ^*^	0.006
HGR [pts]	45.4	6.85	49.8	9.01	−2.59 ^*^	0.011
BAH [pts]	52.5	8.60	52.7	8.23	−0.14 ^*^	0.893
SHR [pts]	55.8	7.57	60.8	8.21	−2.98 ^*^	0.004
SUP [pts]	52.2	8.29	55.2	6.71	−2.16 ^**^	0.031
SAR [pts]	50.9	6.01	49.2	5.62	−1.88 ^**^	0.060

Overall fitness index (OF), running speed (50 mR), standing long jump test (SBJ), long run test (LOR), handgrip strength test (HGR), bent arm hang test (BAH), shuttle run test (SHR), sit-ups in 30 s test (SUP), sit and reach test (SAR), M = mean, SD = standard deviation, ^*^*t* = Student t-test value, ^**^Z = Mann−Whitney U test value, *p* = level of significance.

**Table 3 ijerph-17-05733-t003:** Results of the Table Tennis Specific Battery Test (TTSBT): comparison by gender and training stages.

Variable	Boys		Girls		U Manna-Whitneya Test	
	M	SD	M	SD	Z	*p*
Test 1	13.4	3.32	13.6	3.08	−0.34	0.734
Test 2	13.1	3.91	13.4	3.17	−0.40	0.686
Test 3	10.9	3.03	10.4	3.05	−0.68	0.496
	targeted stage		specialized stage		*Z*	*p*
Test 1	12.5	3.16	14.8	2.77	−3.28	0.001
Test 2	12.0	3.52	14.9	2.98	−3.96	<0.001
Test 3	9.8	3.09	11.9	2.54	−3.47	0.001

M = mean, SD = standard deviation, Z = Mann−Whitney U test value, *p* = level of significance.

**Table 4 ijerph-17-05733-t004:** Correlations between special and general physical fitness.

Variable	Test 1		Test 2		Test 3	
	*r_s_*	*p*	*r_s_*	*p*	*r_s_*	*p*
OF	0.26	0.016	0.24	0.026	0.18	0.101
50 mR [pts]	0.26	0.014	0.23	0.034	0.19	0.084
SBJ [pts]	0.16	0.129	0.24	0.024	0.13	0.215
BAH [pts]	0.32	0.003	0.36	0.001	0.24	0.027
SHR [pts]	0.18	0.095	0.29	0.006	0.12	0.263
SAR [pts]	0.21	0.056	0.21	0.052	0.23	0.029
LOR [pts]	0.11	0.326	0.20	0.057	0.11	0.311
HGR [pts]	0.18	0.089	−0.01	0.953	0.19	0.072
SUP [pts]	−0.04	0.741	−0.06	0.578	−0.03	0.775

Overall fitness (OF) index, running speed (50 mR), standing long jump test (SBJ), bent arm hang test (BAH), shuttle run test (SHR), sit and reach test (SAR), long run test (LOR), handgrip strength test (HGR), sit-ups in 30 s test (SUP), Spearman rank correlation coefficient (*r_s_–); p* = level of significance.

**Table 5 ijerph-17-05733-t005:** Descriptive statistics for physical activity and sedentary behavior.

Variable	Boys		Girls		U Mann−Whitney Test		
	M	SD	M	SD		Z	*p*
MVPA (days)	5.6	1.50	5.3	1.77		−0.75	0.453
VPA (days)	4.2	1.01	4.0	0.87		−0.81	0.420
SB (school/weekend days)	3.0/4.6	1.67/2.25	1.8/2.5	1.35/1.31		−3.52/−4.72	<0.001

M = mean, SD = standard deviation, Z = Mann−Whitney U test value, *p* = level of significance; sedentary behavior (SB).

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
