# Peer review of "The Motor and Leisure Time Conditioning of Young Table Tennis Players’ Physical Fitness"

_ijerph, 2020, doi:10.3390/ijerph17165733_

Round 1
Reviewer 1 Report
General comments
This paper discusses the relationship between the physical fitness of young table tennis players and the determinants of their lifestyle. The authors mentioned that the highlight of this paper is the research on the relationship between physical fitness and arrangement of leisure time in young table tennis players, However, both in the research results and in the discussion part, the research highlights are not well presented. The authors mentioned that ” The correlation analysis showed no statistically significant relationship between the special fitness area (TTSBT) and the leisure time activity of the table tennis players.( line 230) ”, Can the authors explore the relationship between TTSBT and the ratio of physical activity with sedentary in leisure time?
Other comments
1 Abstract It was mentioned in the abstract that the subjects were 9-12 years old (line 11), however, it was mentioned in text that there were two stages include “9-12” and “13-16” (line 80).
2 Participants “young Polish table tennis players from the teams of two provinces: 77
DolnoĹ›lÄ…skie and Wielkopolskie ” (line 77), Why only choose these two provinces?
3 results line 182, are the data of BMI right?
Line 189, The p value should be 0.003, not 0.036.
Author Response
Dear Reviewer,
In responses to your comments in the review of the manuscript "The motor and leisure time conditioning of young table tennis players' physical fitness", we would like to inform you about the undertaken corrections and completions.
We appreciate your thoughtful comments and hope that this revision will be considered much stronger as a result. The authors’ responses are given separately to each Reviewer’s comment. We have tried to address your concerns as fully as possible. Below are the changes that we have introduced.
Thank you for your consideration and we look forward to hearing from you soon.
Reviewer 1.
Authors: First of all, we would like to express our gratitude to Reviewer 1 for the time taken to review our paper and for providing us with comments/suggestions helpful to improve the quality of this paper. We would like to thank for the Reviewer’s valuable contributions. We have found the Reviewer’s criticism and recommendations positive and very constructive. The changes to the paper have been made using red font in the new version of the paper.
General comments
This paper discusses the relationship between the physical fitness of young table tennis players and the determinants of their lifestyle. The authors mentioned that the highlight of this paper is the research on the relationship between physical fitness and arrangement of leisure time in young table tennis players, However, both in the research results and in the discussion part, the research highlights are not well presented. The authors mentioned that ” The correlation analysis showed no statistically significant relationship between the special fitness area (TTSBT) and the leisure time activity of the table tennis players.( line 230) ”, Can the authors explore the relationship between TTSBT and the ratio of physical activity with sedentary in leisure time?
We would like to thank the Reviewer for their valuable remark. The current study did not examine other potentially influential factors, such as access to sport facilities, family traditions in active leisure time and role modelling, as well as school policies, which might be considered a limitation. We mentioned about it in the end of Discussion section. However, similar cultural backgrounds of the examined young table tennis players (coming from urban areas of cities) seem to be a strength. Nevertheless, in further research, multi-contextual and longitudinal studies are needed to consider other potentially mediating variables, and the ratio of physical activity with sedentary in leisure time. These issues will certainly be carefully considered in the next studies.
Other comments:
- Abstract It was mentioned in the abstract that the subjects were 9-12 years old (line 11), however, it was mentioned in text that there were two stages include “9-12” and “13-16” (line 80).
We would like to apologize, it was an unintentional mistake. The age distribution of the examined players ranged from 11 to 17 years. Proper changes were made to the Abstract.
- Participants “young Polish table tennis players from the teams of two provinces: 77
DolnoĹ›lÄ…skie and Wielkopolskie ” (line 77), Why only choose these two provinces?
In both provinces, table tennis is a very popular form of physical activity among young people. This is evidenced by, among others the number of competitor licenses issued in both provinces by the Polish Table Tennis Association (data available at https://www.pzts.pl/licencje/licencje-zawodnicze). Moreover, young players belong to the regional team of both provinces. The authors had direct access to this group of respondents because one of the co-authors of this paper is a coordinator of fitness tests of players from Greater Poland and Lower Silesia.
- results line 182, are the data of BMI right?
BMI was calculated according to the Cole procedure, which was described in Procedures section. Growth references for height, weight, and BMI were constructed with the lambda, mu, sigma (LMS) method using data from a large, recent population-representative sample of school-aged children and adolescents in Poland [Cole TJ The LMS method for constructing normalized growth standards. Eur J Clin Nutr 1990; 44:45–60; KuĹ‚aga Z, RóĹĽdĹĽyĹ„ska-ĹšwiÄ…tkowska A, Grajda A. et al. Percentile charts for growth and nutritional status assessment in Polish children and adolescents from birth to 18 year of age. Standardy Medyczne/Pediatria, 2015; 12: 119-135]. Participants were classified as underweight, normal weight, or overweight according to age- and gender-specific cut-off points for children and adolescents [Cole TJ, Lobstein T. Extended international (IOTF) body mass index cut-offs for thinness, overweight and obesity. Pediatr Obes, 2012; 7(4): 284–294]. The BMI references were based on a current, nationally representative sample of Polish children and adolescents without known disorders affecting growth.
- Line 189, The p value should be 0.003, not 0.036.
The reviewer’s proposal was taken into account.

Reviewer 2 Report
I consider the research of relevance in a very special moment for the dedication of young people in sport. In this global sport, such as tennis, it is important to take into account the different parameters that influence early ages and as talent detection in this sport. I consider that with a small review of English it should be published.
Author Response
Dear Reviewer,
In responses to your comments in the review of the manuscript "The motor and leisure time conditioning of young table tennis players' physical fitness", we would like to inform you about the undertaken corrections and completions.
We appreciate your thoughtful comments and hope that this revision will be considered much stronger as a result. The authors’ responses are given separately to each Reviewer’s comment. We have tried to address your concerns as fully as possible. Below are the changes that we have introduced.
Thank you for your consideration and we look forward to hearing from you soon.
Reviewer 2.
Authors: First of all, we would like to express our gratitude to Reviewer 2 for the time taken to review our paper.
I consider the research of relevance in a very special moment for the dedication of young people in sport. In this global sport, such as tennis, it is important to take into account the different parameters that influence early ages and as talent detection in this sport. I consider that with a small review of English it should be published.
The text of the paper was additionally verified by an English native speaker from Cambridge Proofreading LLC. Therefore, the Authors leave the original version of the indicated text.
Reviewer 3 Report
First, I want to say that this is a great study. I really appreciate the work you all put into it. It was very interesting to read and I believe could be very beneficial for your audience. I did have just a few recommendations.
Line 33: add a comma after "agility [10].
Line 51: I suggest adding "participants" or "players" after "...child and youth table tennis."
Methods/Special Motor Fitness: I would recommend providing slightly more description of this assessment, including the duration of each assessment and elaborating on what was considered as "successful".
2.4 Motor Fitness: I would recommend stating if all participants completed the 8 components in the same order or if it was randomized and the amount of time provided between each component. Was it consistent for everybody? Did they all complete the assessment at the same time so they could compete against each other, or was it completed individually?
2.5 Lifestyle: Did you ask what type of activity they participated in, in addition to the intensity level? Were the participants in school during the study? I would recommend stating if they were or were not because it could effect activity levels.
Line 243: You stated the participants had an average OF score. What are the qualifications/ranges for below average, average, and above average?
- Is there a range for each individual variable?
Author Response
Dear Reviewer,
In responses to your comments in the review of the manuscript "The motor and leisure time conditioning of young table tennis players' physical fitness", we would like to inform you about the undertaken corrections and completions.
We appreciate your thoughtful comments and hope that this revision will be considered much stronger as a result. The authors’ responses are given separately to each Reviewer’s comment. We have tried to address your concerns as fully as possible. Below are the changes that we have introduced.
Thank you for your consideration and we look forward to hearing from you soon.
Reviewer 3.
Authors: First of all, we would like to express our gratitude to Reviewer 3 for the time taken to review our paper and for providing us with comments/suggestions helpful to improve the quality of this paper. We would like to thank for the Reviewer’s valuable contributions. We have found the Reviewer’s criticism and recommendations positive and very constructive. The changes to the paper have been made using red font in the new version of the paper.
First, I want to say that this is a great study. I really appreciate the work you all put into it. It was very interesting to read and I believe could be very beneficial for your audience. I did have just a few recommendations.
Line 33: add a comma after "agility [10].
The lines have been revised as suggested by the Reviewer.
Line 51: I suggest adding "participants" or "players" after "...child and youth table tennis."
The reviewer’s proposal was taken into account.
Methods/Special Motor Fitness: I would recommend providing slightly more description of this assessment, including the duration of each assessment and elaborating on what was considered as "successful".
In the opinion of the authors, section Special motor fitness presented an in-depth description of the test trials. A detailed description of the Table Tennis Specific Battery Test (TTSBT) is included in the positions cited in the manuscript: Gomes F., Amaral F., Venture A., & Agular, J. (2000). Table Tennis specific test battery, International Journal of Table Tennis Sciences, 4, 11-18; Katsikadelis, M., Pilianidis, T., & Mantzouranis, N. (2014). Test-retest reliability of the “table tennis specific battery test” in competitive level young players. European Psychomotricity Journal, 6(1), 3-11.
2.4 Motor Fitness: I would recommend stating if all participants completed the 8 components in the same order or if it was randomized and the amount of time provided between each component. Was it consistent for everybody? Did they all complete the assessment at the same time so they could compete against each other, or was it completed individually?
The change was made as suggested by the Reviewer. Additional information was introduced to confirm the content of the IPFT tests. The athletes started the tests after they had performed standardized warm-up general activities (15 minutes). We would like to confirm that all players did 8 test runs in the same order. This is in accordance with the test requirements. The standardization of the test parameters is captured in protocols, which include a detailed description of materials, set-up, assignment, demonstration, training phase, testing phase and registering test scores. All participants performed their tests on the same day during a sports camp in standardized conditions.
2.5 Lifestyle: Did you ask what type of activity they participated in, in addition to the intensity level? Were the participants in school during the study? I would recommend stating if they were or were not because it could effect activity levels.
We would like to thank the Reviewer for their valuable remark. The information about the questionnaire survey procedure was properly introduced to the Lifestyle section. We asked table tennis players about their favourite type of additional physical activity (PA). The analysis of the collected data showed that the most common forms of PA undertaken in leisure time were walking and cycling. Participants were not at school during the research. All respondents were informed of the aim of the research, the manner in which the results would be used for scientific purposes, and that their participation would be anonymous. Questionnaires were completed in whole groups during sports camp and took approximately 25 minutes to complete.
Line 243: You stated the participants had an average OF score. What are the qualifications/ranges for below average, average, and above average?
Is there a range for each individual variable?
Overall physical fitness was determined using the International Physical Fitness Test (IPFT). The classification standard for each of the individual IPFT trials is 50 points. In Poland, pursuant to the guidelines of the Polish Ministry of Sport and Tourism, the scores in this test are one of the main criteria used to appoint members of regional teams in various sports. IPFT scores are used to monitor progress in training of athletes and are included in the annual reports prepared for the Competitive Sports Department. For the sake of evaluating general fitness of a participants (OF), the points obtained after each test should be added up. It is more convenient to divide the sum of points by the number of tests performed, because this yields the average level of general physical fitness of the individual on a 1-100 scale – that is, in the same scale as for each particular test. The score tables are presented in the studies:
- Pilicz S, Charzewski J, eds. Punktacja sprawności fizycznej młodzieży polskiej wg Międzynarodowego Testu Sprawności Fizycznej. Kryteria pomiaru wydolności organizmu testem Coopera. AWF, Warszawa 2004 (http://www.zs9.bydgoszcz.pl/sites/default/files/MTSF%20opis%20AWF.pdf).
- Przewęda R., Dobosz J. (2006). The physical fitness of Polish youth. Studia i Monografie 98. Warsaw: AWF Warsza-wa.
- Rosandich TP. International physical fitness test. Sport J. 1999; 2.
The score tables presented in this study serve to gauge in terms of points the results of eight exercise tests composing the test developed by the International Committee on the Standardisation of Physical Fitness Tests. The score tables facilitate evaluation or self-evaluation of the level of pupils’ physical fitness in comparison to other pupils – not necessarily peers – as well as in comparison to their own results from previous years.

Reviewer 4 Report
The authors conducted a study on what factors influence leisure activity on the fitness level of young table tennis players.
An interesting approach has been attempted in which young players' choice of leisure time can influence their future growth as a global player.
This study was limited to table tennis and was conducted in correlation between the leisure time (lifestyle) and the physical fitness of young players, but if more data were analyzed in the relationship with other similar sports, it could contribute to the development of sports science.
In particular, if a suitable leisure activity for the player is suggested in each sport, it is expected to be of great help in developing their potential.
Therefore, it is recommended that “Accept after minor revision”
Since there are many abbreviations in the manuscript, it is recommended to add summary of abbreviations (nomenclature) appropriate to the format of this journal.
Typos
Line 1: conditioningof young -> conditioning of young
Line 135: 2.5Lifestyle -> 2.5. Lifestyle
References should be carefully revised as the journals format.
A doi number or reference link should be provided after each reference.
Line 364: underlines
Line 366: 36(23),2716-2723 -> 36(23), 2716-2723
Line 386,
Line 396,
Line 402: 4(1-1),15-21. -> 4(1-1), 15-21.
Line 416,
Line 448: vol, page
Line 458: et al.
Line 462,
Line 463: page
Line 467: 2016a, 28(4):553-64. -> 2016, 28(4), 553-64.
Line 470: 2016b
Line 494: 2014,48 -> 2014, 48
Line 518: page
Author Response
Dear Reviewer,
In responses to your comments in the review of the manuscript "The motor and leisure time conditioning of young table tennis players' physical fitness", we would like to inform you about the undertaken corrections and completions.
We appreciate your thoughtful comments and hope that this revision will be considered much stronger as a result. The authors’ responses are given separately to each Reviewer’s comment. We have tried to address your concerns as fully as possible. Below are the changes that we have introduced.
Thank you for your consideration and we look forward to hearing from you soon.
Reviewer 4.
Authors: First of all, we would like to express our gratitude to Reviewer 4 for the time taken to review our paper and for providing us with comments/suggestions helpful to improve the quality of this paper. We would like to thank for the Reviewer’s valuable contributions. We have found the Reviewer’s criticism and recommendations positive and very constructive. The changes to the paper have been made using red font in the new version of the paper.
The authors conducted a study on what factors influence leisure activity on the fitness level of young table tennis players. An interesting approach has been attempted in which young players' choice of leisure time can influence their future growth as a global player. This study was limited to table tennis and was conducted in correlation between the leisure time (lifestyle) and the physical fitness of young players, but if more data were analysed in the relationship with other similar sports, it could contribute to the development of sports science. In particular, if a suitable leisure activity for the player is suggested in each sport, it is expected to be of great help in developing their potential.
Therefore, it is recommended that “Accept after minor revision”
Since there are many abbreviations in the manuscript, it is recommended to add summary of abbreviations (nomenclature) appropriate to the format of this journal.
The summary of abbreviations was introduced as suggested by the Reviewer (line: 371-378).
Typos
Line 1: conditioningof young -> conditioning of young
The line has been revised as suggested by the Reviewer.
Line 135: 2.5Lifestyle -> 2.5. Lifestyle
The line has been revised as suggested by the Reviewer.
References should be carefully revised as the journals format.
A doi number or reference link should be provided after each reference.
The lines have been revised as suggested by the Reviewer, doi numbers were added (where available).
Line 364: underlines - done
Line 366: 36(23), 2716-2723 -> 36(23), 2716-2723 - done
Line 386, - done
Line 396, - done
Line 402: 4(1-1),15-21. -> 4(1-1), 15-21. - done
Line 416, - done
Line 448: vol, page – We added: Article ID – 5489348 and number of 8 pages
Line 458: et al. - This document has a large number of co-authors (more than 10 authors), so we cited the first ten authors, then added a semicolon and add ‘et al.’ at the end.
Line 462, - done
Line 463: page - done
Line 467: 2016a, 28(4):553-64. -> 2016, 28(4), 553-64. - done
Line 470: 2016b- done
Line 494: 2014,48 -> 2014, 48- done
Line 518: page- done

Reviewer 5 Report
The manuscript is well written and takes into account the missing parameters to gauge performance and health analysis of young tennis players. However, it will be interesting to analyse how these change in different age groups. Here are a few comments addressing which would strengthen the research work:
- In the title, spacing between 'conditioning of' is missing.
-
In the introduction, 1st sentence (line 24) could be rephrased to avoid saying complex game.
-
In line 28, it would be appropriate to define why it is anaeobic metabolism to set the context.
-
It would be better to cite a few references for the statement made in Line 51 and 52 as to why this rationale is interesting.
-
In line 54, start the sentence with Young instead of 'youth'
-
In line 58, using the term 'parameters' would be more scientific then using 'items'.
-
Reference missing for Line 62,63 sentence.
-
Reference for Line 67,68,69 is missing.
-
In line 112, it is not mentioned what ICC means.
-
In line 113, it might be good to explain what Cronbach's alpha is or measures.
-
in line 134, it is unclear how OF index was calculated.
-
It would be appropriate to include a summary flowchart/schematic of the tests performed/proposed in this study as a tool kit for application by the users.
Author Response
Dear Reviewer,
In responses to your comments in the review of the manuscript "The motor and leisure time conditioning of young table tennis players' physical fitness", we would like to inform you about the undertaken corrections and completions.
We appreciate your thoughtful comments and hope that this revision will be considered much stronger as a result. The authors’ responses are given separately to each Reviewer’s comment. We have tried to address your concerns as fully as possible. Below are the changes that we have introduced.
Thank you for your consideration and we look forward to hearing from you soon.
Reviewer 5.
Authors: First of all, we would like to express our gratitude to Reviewer 5 for the time taken to review our paper and for providing us with comments/suggestions helpful to improve the quality of this paper. We would like to thank for the Reviewer’s valuable contributions. We have found the Reviewer’s criticism and recommendations positive and very constructive. The changes made to the paper have been made using red font in the new version of the paper.
The manuscript is well written and takes into account the missing parameters to gauge performance and health analysis of young tennis players. However, it will be interesting to analyse how these change in different age groups. Here are a few comments addressing which would strengthen the research work:
In the title, spacing between 'conditioning of' is missing.
The line has been revised as suggested by the Reviewer.
In the introduction, 1st sentence (line 24) could be rephrased to avoid saying complex game.
The Reviewer’s remark was taken into account.
In line 28, it would be appropriate to define why it is anaeobic metabolism to set the context.
We would like to thank the Reviewer for the valuable remark. The information about the physiological profile of racket sports (especially about table tennis) was properly introduced to the Introduction section. We mentioned about it in lines 43 - 47.
It would be better to cite a few references for the statement made in Line 51 and 52 as to why this rationale is interesting.
It is difficult to indicate specific references for this statement. The authors wanted to indicate this direction of research as important for young table tennis players, based on their own coaching experience. One such reference was quoted in the paper (Exel et al., 2018).
In line 54, start the sentence with Young instead of 'youth'
The line has been revised as suggested by the Reviewer.
In line 58, using the term 'parameters' would be more scientific then using 'items'.
The line has been revised as suggested by the Reviewer.
Reference missing for Line 62,63 sentence.
Additional references were properly introduced to the sentence.
Reference for Line 67,68,69 is missing.
The change was made as suggested by the Reviewer. Additional references were properly introduced to these lines.
In line 112, it is not mentioned what ICC means.
ICC - intraclass correlation coefficient – the explanation was added.
In line 113, it might be good to explain what Cronbach's alpha is or measures.
Cronbach’s alpha is a measure of internal consistency, that is, how closely related a set of items are as a group. It is considered to be a measure of scale reliability. The explanation was added.
In line 134, it is unclear how OF index was calculated.
The International Physical Fitness Test consists of eight exercise tests. For the sake of evaluating general fitness of a participants (OF), the points obtained after each test should be added up. It is more convenient to divide the sum of points by the number of tests performed, because this yields the average level of general physical fitness of the individual on a 1-100 scale – that is, in the same scale as for each particular test. The score tables are presented in the study - Pilicz S, Charzewski J, eds. Punktacja sprawnoĹ›ci fizycznej mĹ‚odzieĹĽy polskiej wg MiÄ™dzynarodowego Testu SprawnoĹ›ci Fizycznej. Kryteria pomiaru wydolnoĹ›ci organizmu testem Coopera. AWF, Warszawa 2004 (http://www.zs9.bydgoszcz.pl/sites/default/files/MTSF%20opis%20AWF.pdf).
It would be appropriate to include a summary flowchart/schematic of the tests performed/proposed in this study as a tool kit for application by the users.
The figure illustrated recommended tests to gauge motor and leisure activity of young tennis players was added.

Round 2
Reviewer 1 Report
The paper can be published with the author's modification.
Author Response
We would like to thank the Reviewer for taking the time to make a thorough revision to our manuscript and provide constructive comments. We feel that the review has helped to enhance the quality of the article.